# Interaction between Sound and Thermal Influences on Patient Comfort in the Hospitals of China's Northern Heating Region

**Yue Wu, Qi Meng \*, Lei Li and Jingyi Mu**

Key Laboratory of Cold Region Urban and Rural Human Settlement Environment Science and Technology, Ministry of Industry and Information Technology, School of Architecture, Harbin Institute of Technology, Harbin 150001, China; wuyuehit@hit.edu.cn (Y.W.); ks980313@gmail.com (L.L.); mujingyi1713@163.com (J.M.)
\* Correspondence: mengq@hit.edu.cn; Tel.: +86-115104515125



**Featured Application: Authors are encouraged to provide a concise description of the specific application or a potential application of the work. This section is not mandatory.**

**Abstract:** Previous studies have found that hospitals are often inadequately ventilated in the heating region of China, which causes an increased risk of negative impacts on patients. The complex interaction between thermal comfort and acoustics presents considerable challenges for designers. There is a wide range of literature covering the area of the interaction between the sound–thermal, sound–odor, and acoustic–visual influences, but a focused research on the sound –thermal influence on comfort in hospitals has not been published yet. This paper describes a series of field measurements and subjective evaluations that investigate the thermal comfort and acoustic performance of eighteen hospitals in China. The results showed that the thermal comfort in the monitored wards was mostly acceptable, but the temperatures tended to be much higher and the humidity much lower, in practice than they were designed to be in the heating season. The most significant conclusion is that a positive thermal stimulus can create a comfortable thermal environment, which can improve patients' evaluation of the acoustics, while a negative stimulus has the opposite effect. A comfortable acoustic environment also caused patients to positively evaluate thermal comfort. Moreover, the relationship between thermal and sound effects in the overall evaluation showed that they are almost equal.

**Keywords:** patient satisfaction; thermal comfort; acoustic comfort; heating region; hospitals

## 1. Introduction

In complex and diverse healthcare environments, environmental comfort is a key factor that affects the evaluation of patient satisfaction [1]. The acoustic and thermal environment has the most direct influence on the patient's comfort level and satisfaction. Currently, there are a considerable number of thermal comfort studies on hospitals and other healthcare buildings. Some studies are focused on environmental parameters, such as the indoor temperature, humidity, and air movement [2], while some other investigations have been presented in terms of the thermal discomfort and thermal sensation of patients and hospital staff [3–5]. According to the international standards of hospitals, the desirable indoor air temperature of regular wards is 20–24 °C, and the recommended levels of relative humidity are from 30% to 60% (ISO 7330 [6]). Previous research has shown that a high temperature may cause an increased out-gassing of toxins from building materials and provide more favorable growing conditions for bacteria, and low humidity can increase susceptibility to respiratory disease and contribute to irritation [7]. Therefore, the standards for temperature and humidity ranges

in healthcare buildings are influenced by the measure of infection control. However, whether the thermal environment within this range is comfortable for patients seems to be ignored.

Acoustic comfort in hospital buildings has also been widely studied by researchers. Noise has been identified as a major stressor in hospitals [8]. Qin et al. [9] proposed that the general noise from nearby people was the most annoying sound, and patients' social characteristics would not significantly affect their acoustic environment evaluation. Different types of noise can be considered as acoustic pollution, and patients exposed to high amounts of noise are at a much higher risk [10]. Researchers have also measured the noise levels or studied the sound field of various healthcare environments [11,12]. Xie and Kang found that the hospital acoustic environment differed significantly every night; meanwhile, more intrusive noises tended to originate from multi-bed wards, while more extreme sounds were likely to occur in single wards [13]. Studies have found that long-term exposure to noise can lead to a range of health problems, such as an increase of cardiovascular morbidity [14] and blood pressure [15], sleep disturbances [16], and overproduction of cortisol [17]. However, the sound is not always annoying [18]. Thomas et al. showed, through a survey of a nursing home, that acoustic interventions had direct positive outcomes, as well as both positive and negative outcomes in terms of the perceived indirect effects [19].

Patients' feeling of comfort is influenced by a variety of environmental factors, and some researchers have conducted environmental interaction research. Humans perceive comfort through the interaction of various sensory stimuli and their integration in various environments. Thermal and acoustic influences are the two aspects that have the greatest influence on environmental comfort [20]. Previous research has shown that thermal comfort and acoustic comfort may influence each other [21]. A research on schools, as a case study, found that acoustic comfort was affected by thermal factors, especially in winter (Mumovic et al., 2009) [22]. Pellerin and Candas (2004) proposed that acoustic comfort and perception were affected by temperature [23]. Yang and Moon (2019) found that the impact of acoustics on indoor environmental comfort was the greatest and acoustic comfort increases under thermo neutrality [24]. However, little attention has been paid to the interactions between the sound and thermal influences inside hospitals. Meanwhile, all of these case studies were undertaken in just a single hospital, and general conclusions based on the results from just one case study are considered to be insufficient.

Therefore, the aim of the paper is to discern the interaction between the thermal and acoustic influences on environmental comfort in China's northern heating region. The general wards of 18 hospitals were investigated, and measurements of environmental factors were performed, including thermal and acoustic parameters. Thermal and acoustic evaluations were also carried out through survey questionnaires and purposely elaborated. Finally, the interaction between thermal and acoustic factors was discussed.

## 2. Methodology

### 2.1. Measurements

In China's heating region, the annual average daily temperature is at a stable ≤5 °C for over 90 days. The area of the heating region constitutes 70% of China's land area. The construction area of this region is about 6.5 billion m$^2$ [25]. In this paper, our investigation focuses mainly on the northern heating region of China, including the three northeastern provinces and the Eastern Inner Mongolia autonomous region, which accounts for 51.6% of the heating region. As case studies, 18 hospitals were chosen from six cities in China's northern heating region, including Harbin, Wuchang, Qitaihe, Chifeng, Changchun, and Meihekou. The measured parameters and instruments are shown in Table 1.

**Table 1.** Measured indoor and outdoor thermal comfort parameters, operative range, and accuracy.

| Parameters | Definition | Measurement System | Instrument |
|---|---|---|---|
| T (°C) | forced airy dry-bulb air temperature | ISO 7726 | Testo 435 |
| RH (%) | relative humidity | ISO 7726 | Testo 435 |
| LAeq (dB) | equivalent A-weighted sound pressure level | ISO 3382 | BSWA801 |

As for the measurement of the indoor environment, the measurement time for the current study was 8:00–18:00. Thus, it can be guaranteed that the participants will not be disturbed by activities, such as ward rounds, family visits, dining, etc., which can lead to crowd movement and affect the physical environment. These instant measurements included the temperature, relative humidity, and sound pressure level in particular hospital wards to record the environmental data while the participant responded to the survey questionnaire.

The thermal environment of the hospitals was measured from 15 November to 30 February 2018. This period provides heating for the winter in these cities. Due to the low outdoor temperature and lack of natural ventilation during heating, the airflow rate in the room is very low, so the effect of the airspeed was not considered. The air temperature and relative humidity were measured in the wards in the hospitals with Testo 435 thermo-recorders. The measuring devices were placed on the desk (0.6 m above the floor) to record the thermal environment experienced by the respondent at that time. All measurement intervals were 10 min.

To measure the sound environment, the equivalent continuous A-weighted sound pressure level (LAeq) was immediately recorded using BSWA 801 sound-level meters. During the measurement, the sound level meters were set in slow-mode and A-weight, and reading was acquired every 3–5 s. A total of 5 min of data were obtained at each survey position [26]. Additionally, the distance between the measurement location, walls, and other major reflective surfaces was ensured to be at least 1 m, and the distance between the measurement location and the ground was 1.2–1.5 m [27]. To avoid sound source variability, each sound pressure level at each measurement point was tested 10 times; each measuring point was tested every hour, and the average value of the 10 sets of data was taken as the result of this measurement point. The measuring period lasted from 8:00 to 18:00. The equipment selection and measurement process followed the ISO3382 standard. A mean value was calculated to obtain the spatially averaged LAeq value [28].

*2.2. Questionnaire Survey*

The participants were patients at large general hospitals in China's northern heating region. The project only deals with the evaluation of the hospital's building environment. It does not involve the investigation of the patient's condition and does not increase the pain and psychological impact of the patient's treatment process. This study does not belong to biomedical research. An ethics review and approval for this study was not required according to China's ethics, as stated in the Methods for ethical review of biomedical research involving human beings" [29], from the Harbin Institute of Technology's guidelines and national regulations. The research involved in the study and the plan fully considered the principles of safety and fairness. After the professor's associations from the school of architecture of the Harbin Institute of Technology reviewed the study, the rights and interests of the respondents were fully protected and the research content was guaranteed not to cause harm and risk to the respondents. The recruitment of respondents was based on the principles of voluntary and informed consent, and the subjects' rights and privacy were protected. All participants gave written informed consent. There was no conflict of interest between the research content and the research results. The survey process and questionnaire were conducted by the first author (YW), a postdoctoral student with many years of experience conducting research interviews. Finally, the questionnaire was agreed upon by the expert panel, as shown in Table 2. The interview and research work of the project was approved to be conducted as planned. The sample comprised 220 participants (110 males and 110 females), with an average age of 49 (SD = 15.01).

**Table 2.** Questionnaire questions and scales.

| Questions | Scale |
| --- | --- |
| Gender | 1, male; 2, female |
| Age | 1, <20; 2, 20–40; 3, 41–60; 4, >60 |
| The duration of stay | 1, <2 days; 2, 2–5 days; 3, 6–10 days; 4, 11–20 days; 5, >20 days |
| The number of beds | 1; 2; 4; 6; 8 |
| Evaluation of the overall environmental comfort | Five-point scale: 1 very uncomfortable–5 very comfortable |
| Do you agree that the thermal environment is satisfactory? | Five-point scale: 1 strongly disagree–5 strongly agree |
| Do you agree that the temperature is satisfactory? | Five-point scale: 1 strongly disagree–5 strongly agree |
| Do you agree that the humidity is satisfactory? | Five-point scale: 1 strongly disagree–5 strongly agree |
| Acoustic comfort of various sound sources | Five-point scale: 1 very uncomfortable–5 very comfortable |
| Do you agree that the acoustic environment is satisfactory? | Five-point scale: 1 strongly disagree–5 strongly agree |

Participants were interviewed individually and briefed on the purpose of the study, and they gave written informed consent to participate in the research. The interview was usually conducted in the hospital ward, but in a few cases, they were conducted in a separate room for the sake of patient privacy. Ten trained research assistants, recruited from doctor and master candidates, tested the patients. During the interviews, the participants were asked to describe situations, which they thought would bring them satisfaction or dissatisfaction in terms of the thermal and acoustic environment in which they would be treated. Field notes were written after each interview to document the immediate responses to the interaction that had occurred and aid reflexivity.

*2.3. Statistical Analyses*

The data that we obtained in our experiment were analyzed with SPSS 15.0 [30]. The Spearman correlation was used to determine the factors affecting patients' comfort evaluation of the overall environment, and the mean differences (*t*-test, two-tailed) in the influence of these factors on patients' comfort. The Spearman correlation analysis and regression analysis were used to determine the different building environment factors affecting patient comfort. One-way ANOVA was adopted to ascertain the factors affecting patients' comfort evaluation, including demographic and social factors, as well as the interaction between thermal and acoustic influences.

## 3. Results

*3.1. Comfort Evaluation*

### 3.1.1. Overall Comfort Evaluation

According to the satisfaction evaluation of the comfort level, it should be noted that the overall comfort of the hospitals was acceptable (mean value was 4.1). The participants mainly reported "comfortable" (62.3%) and "very comfortable" (25%). However, 12.7% of the participants insisted that the satisfaction level was unacceptable (scale ≤3). Table 3 shows the mean and standard deviation of the patients' physical environment satisfaction evaluation. It can be seen, from column A, that the comfort of the individual physical environment factors was acceptable. The patients' satisfaction with the acoustic environment was highly acceptable (with a mean of value 4.40).

An analysis using Spearman correlation between the physical environment subjective evaluation and overall comfort was conducted, and the correlation coefficient was shown in Table 3 (see column B) ($p < 0.01$). The results showed that there was a significant positive correlation between the satisfaction evaluation of the temperature and acoustic with the satisfaction evaluation of the overall environmental comfort, and the temperature is more relevant than acoustics. The correlation between humidity and overall environmental satisfaction is not significant. In order to discuss the specific effects of these factors on patient comfort, the following part focuses on the influence of various individual physical environmental factors on patients' comfort evaluation.

**Table 3.** Mean and standard deviation (SD) of the satisfaction with the physical environment and a Spearman correlation analysis of the physical environment factors and the overall comfort evaluation.

| Type of Physical Environment Factors | A<br>Mean and Standard Deviation SD of Physical Environment Satisfaction | B<br>Correlation Coefficient between the Factors and Overall Comfort |
|---|---|---|
| Temperature | 4.04/0.911 | 0.686 ** |
| Humidity | 4.10/0.661 | 0.302 |
| Acoustic | 4.40/0.637 | 0.618 ** |

** Correlation is significant at the 0.01 level (2-tailed).

### 3.1.2. Thermal Evaluation

The mean indoor air temperature of the wards during the heating period was approximately 28 °C, the temperature was kept between 25 °C and 31 °C in the wards, and the relative humidity was kept between 20% relative humidity (RH) and 50% RH. Taking the measurement results and subjective evaluation of the temperature and humidity, as shown in Figure 1, it should be noted that the temperature in the wards was satisfactory, with a measured value of 26–30 °C, and highly acceptable at 27–29 °C. The patients had a preference for temperatures that are higher than the standard value in ISO 7730 and ASHRAE 55-2004 [31,32]. Apart from the environmental parameters, two personal variables, i.e., activity and clothing level of the occupants are also very important [33]. Usually, in the ward, the patient is less active and lies in bed for a long time. In addition, the clothing is thinner, so their thermal sensation will be significantly higher than the standard comfort values. The relative humidity was unsatisfactory, and participants mainly reported dissatisfied (37.7%) and moderate (28.2%); 12.3% of the participants even claimed that they were very unsatisfied. With mean ambient temperatures around 28 °C, the relative humidity is low, and heating exacerbates this phenomenon.

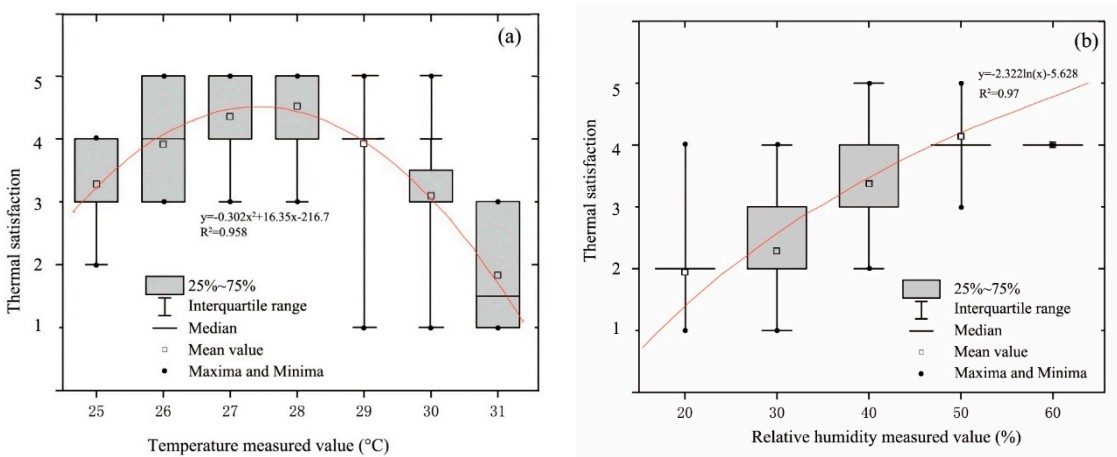

**Figure 1.** The relationship between the measured temperature and relative humidity with thermal satisfaction. (**a**) Temperature, (**b**) relative humidity.

The measurement results of the dominant thermal parameters indicated that they may have different influences on thermal satisfaction and thermal perception. The relationship between the measured temperature and relative humidity with thermal satisfaction are also compared, where the linear regressions and coefficient of determination $R^2$ are also presented. It should be noted that, unlike in previous studies, in China's heating region, patients felt satisfied with an indoor temperature of 26–28 °C and a relative humidity of over 40%. With the increase of the measured temperature, the thermal satisfaction is first increased, and then drops rapidly after 28 °C. The measured temperature and thermal satisfaction constituted a polynomial linear regression, and the coefficient of determination $R^2$ was 0.958. However, the thermal environment satisfaction increased as the relative humidity rose.

The measured relative humidity and thermal satisfaction constituted an exponential linear regression, and the coefficient of determination R$^2$ was 0.97.

### 3.1.3. Acoustic Evaluation

Previous studies suggested that different sound source and behavior patterns influence the acoustic perception of users in an indoor environment [34]. In the survey wards, the sound source composition and behavior patterns are relatively simple. The main sound sources are speech and activity. Table 4 provides a statistical analysis using Spearman's correlation between the acoustic comfort evaluation of various individual sound sources and the comfort evaluation of the overall acoustic environment ($p < 0.01$). The results showed that there was a positive correlation between the acoustic comfort evaluation of the speech sound of staff, shouting, crying, and pager sounds. Shouting and crying were the most relevant. In summary, these types of sounds were very disturbing, and when these types of sound were the dominant sound sources, the ambient acoustic comfort rating would significantly affect the overall comfort.

**Table 4.** Analysis using Spearman's correlation between the acoustic comfort of sound sources and the overall sound environment comfort evaluation.

| Type of Sound Sources | Sound Source Composition | Coefficient Correlation |
|---|---|---|
| Speech sound | Speech of companions | 0.35 * |
| | Chatting of other users | 0.067 |
| | Speech of staff | 0.286 ** |
| | Shouting | 0.438 ** |
| | Phone calls | 0.098 |
| | Crying | 0.426 ** |
| Activity sound | TV sounds | −0.198 * |
| | Pager sounds | 0.392 ** |
| | Walking sounds | 0.061 |

* Correlation is significant at the 0.05 level (2-tailed). ** Correlation is significant at the 0.01 level (2-tailed).

The mean value of the LAeq in the survey wards during the daytime was 59.2 dB, and the LAeq was kept between 57.3 dB and 63.8 dB. As shown in Figure 2, it should be noted that the LAeq in wards was satisfactory, with the measured value of 45–65 dB, and highly acceptable at 45–55 dB. However, when the LAeq reached 70 dB, the participants mainly reported dissatisfied (43.5%) and strongly dissatisfied (21.7%). This result indicated that patients prefer a quieter environment.

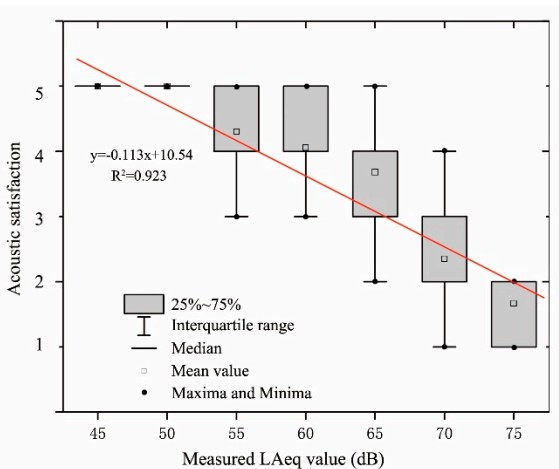

**Figure 2.** The relationship between the measured equivalent continuous A-weighted sound pressure level (LAeq) and acoustic satisfaction.

Figure 2 shows the relationship between the measured LAeq and acoustic satisfaction, where the linear regressions and coefficient of determination $R^2$ are also presented. It should be noted that LAeq lower than 65 dB allowed for a satisfactory acoustic environment. With an increase in the measured LAeq, the acoustic satisfaction decreased. A correlation analysis of the evaluation of the measured value and satisfaction evaluation indicated that evaluation of the satisfaction of the acoustic environment was significantly and negatively correlated with the sound pressure level. The noisier the background noise, the lower the patients' acoustic satisfaction evaluation of the environment.

### 3.2. Interaction between the Thermal and Acoustic Environment

### 3.2.1. The Effect of the Sound and Thermal Stimuli

Table 5 shows the mean rating change of the acoustic comfort evaluation, relative to the condition, with and without thermal stimuli. With the introduction of thermal factors, the mean ratings changed, compared with the condition of no thermal factors. When the temperature was low, the changes of acoustic comfort ratings were mostly less than 0.03; when the temperature was medium and high, acoustic comfort was reduced; and with the increase of temperature, the reduced decline was more obvious. No matter the type of sound source, when the humidity was low, the effect on the acoustic comfort was the most obvious, but when the humidity was medium or high, it was non-significant. In the case of shouting, crying, and pager sounds, the effects of temperature and humidity were more obvious, and high temperature and low humidity had negative influences. However, with a medium temperature and humidity, the effect of improving acoustic comfort got better. In sum, the effect of thermal factors on the acoustic comfort of shouting, crying, and pager sounds was the most evident, and that of speech, TV sounds, and walking sounds was the least evident. Table 5 also shows the influence of measured time on the acoustic comfort evaluation. The occasional sound had no noticeable effect at different times, but the speech sounds and activity sounds show significant differences at different time periods. During the lunch break, for example, there was a significant reduction in these sounds due to human behavior, resulting in a higher score for the acoustic environment.

**Table 5.** Mean rating change of acoustic comfort, relative to the thermal condition.

| Indicator | Sound | Without Thermal Stimuli | Measured Time | | | | | Temperature | | | Humidity | | |
|---|---|---|---|---|---|---|---|---|---|---|---|---|---|
| | | | 8am-10am | 10am-12am | 12am-2pm | 2pm-4pm | 4pm-6pm | Low | Medium | High | Low | Medium | High |
| Speech sound | Speech sound of companions | 0.18 | 0.15 | 0.12 | 0.36 | 0.24 | 0.13 | 0.15 | 0.17 | 0.13 | 0.02 | 0.21 | 0.18 |
| | Chatting sound of other users | 0.11 | 0.09 | 0.07 | 0.23 | 0.18 | 0.08 | 0.07 | 0.09 | -0.21 | 0.13 | 0.08 | 0.11 |
| | Speech sound of staff | 0.22 | 0.12 | 0.13 | 0.39 | 0.26 | 0.24 | 0.23 | 0.26 | 0.11 | 0.09 | 0.21 | 0.19 |
| | Shout | -0.23 | -0.39 | -0.28 | 0.18 | 0.02 | -0.09 | -0.21 | -0.25 | -0.39 | -0.28 | -0.19 | -0.17 |
| | Phone call | -0.11 | -0.12 | -0.19 | -0.05 | -0.08 | -0.17 | -0.08 | -0.09 | -0.17 | -0.13 | -0.07 | -0.05 |
| | Cry | -0.34 | -0.35 | -0.38 | -0.29 | -0.29 | -0.32 | -0.28 | -0.31 | -0.46 | -0.38 | -0.32 | -0.25 |
| Activity sound | TV sound | 0.11 | 0.09 | 0.13 | 0.08 | 0.16 | 0.11 | 0.19 | 0.15 | 0.03 | 0.08 | 0.03 | 0.04 |
| | Pager sound | -0.02 | 0.03 | 0.02 | -0.08 | 0.01 | -0.02 | 0.08 | 0.16 | -0.07 | -0.09 | -0.03 | -0.02 |
| | Walking sound | 0.02 | 0.06 | 0.02 | -0.09 | 0.05 | 0.04 | 0.03 | 0.02 | -0.04 | 0.04 | 0.01 | 0.04 |

### 3.2.2. The Interaction between the Thermal and Acoustic Influences on Comfort

Despite the results presented in the literature [35], where thermal comfort is in general ranked by the building occupants as having a greater importance than visual and acoustic ones, in this survey, the thermal and acoustic factors had about the same weight. Existing research indicated that indoor environments were formed under the combined action of various physical environmental factors [36]. The interaction of these factors should be taken into consideration.

Mixed design analyses of variance (ANOVAs) were run to test the influence of different temperatures and humidity levels on the sound pressure level. As shown in Table 6, the significance level (Sig.) is the *p*-value of the variance F test. $p \leq 0.05$ means that the "temperature" and "humidity" significantly influence the LAeq at a level of 0.05. The deviation means square of the sound pressure level for different temperatures and humidity levels (A * B) is 18.902, the F value is 1.418, and the significance level is 0.298, that is, $p > 0.05$ shows no significant difference. The results of the multiple comparison analysis under the interaction of the sound pressure level and temperature showed that there was no significant difference between 18 °C and 22 °C ($p > 0.05$), as shown in Table 7. The interaction between the LAeq and temperature showed that with the increase of temperature, the LAeq was significantly increased. The reason may be that people experienced more negative emotions when the thermal environment deviated from neutral conditions [37], the volume was automatically increased, or the ratio of shouting and crying was higher.

**Table 6.** Mixed design analyses of variance (ANOVAs). Dependent Variable: LAeq.

| Source | Mean Square | F | Sig. |
|---|---|---|---|
| Temperature (A) | 1575.423 | 90.815 | 0.000 |
| Humidity (B) | 313.087 | 18.512 | 0.000 |
| Temperature * Humidity (A * B) | 18.902 | 1.428 | 0.298 |

**Table 7.** Multiple comparisons analyses. Dependent Variable: LAeq.

| Temperature (I) | Temperature (J) | Mean Difference (I–J) | St. Error | Sig. | 95% Confidence Interval | |
|---|---|---|---|---|---|---|
| | | | | | Lower Bound | Upper Bound |
| 18 | 22 | 6.018 | 1.700 | 0.116 | −1.23 | 13.266 |
| | 26 | 18.176 ** | 1.700 | 0.000 | 12.968 | 23.384 |
| | 30 | 26.897 ** | 1.700 | 0.000 | 17.616 | 36.178 |
| 22 | 18 | −6.018 | 1.700 | 0.116 | −13.266 | 1.23 |
| | 26 | 12.088 ** | 1.700 | 0.000 | 10.665 | 13.511 |
| | 30 | 15.286 ** | 1.700 | 0.000 | 9.683 | 20.889 |
| 26 | 18 | −18.176 ** | 1.700 | 0.000 | −23.384 | −12.968 |
| | 22 | −12.088 ** | 1.700 | 0.000 | −13.511 | −10.665 |
| | 30 | 9.575 ** | 1.700 | 0.002 | 3.981 | 15.169 |
| 30 | 18 | −26.897 ** | 1.700 | 0.000 | −36.178 | −17.616 |
| | 22 | −15.286 ** | 1.700 | 0.000 | −20.889 | −9.683 |
| | 26 | −9.575 ** | 1.700 | 0.002 | −15.169 | −3.981 |

** The mean difference is significant at the 0.01 level.

## 4. Discussion

The factors affecting patients' comfort in hospital wards include factors other than acoustic and thermal influences. In the following paragraphs, some major factors are discussed, which must be reviewed and updated in future studies.

**House-related factors**. Aspects related to the house configuration significantly affect the indoor environment. The room location (facing the street vs. facing a quiet side) was found to be correlated with road traffic noise annoyance [38]. In this study, the room location has a significant influence on acoustic comfort and thermal comfort. Rooms facing the streets were found to negatively affect annoyance caused by the passage of nearby traffic. However, due to these rooms facing south, bringing in more light, indoor thermal conditions were rated relatively higher than those facing north.

Figure 3 provides the measured LAeq over the test period with different ward capacities. Since the density of patients and the size of the space affect the indoor thermal and acoustic environment, we investigated wards with different numbers of beds. It should be noted that the environmental noise in six-bed rooms and eight-bed rooms were higher than that in single, double, and quod rooms. This indicated that the patient density had a significant impact on environmental noise, which is consistent with a study of an underground commercial street [39]. The noise level of each ward starts to rise at 8:00 and becomes relatively quiet from 12:00 to 14:00 because of the lunch break. The peak noise at 70–75 dB occurs from 9:00–11:00 and 15:00–17:00, and this level of noise can be annoying or uncomfortable. As in previous studies, more occurrences of noises with longer duration were observed in multiple-bed wards, as compared with single-bed wards [40]. This indicates that the measurement results of the acoustic environment were similar to those in different climate regions.

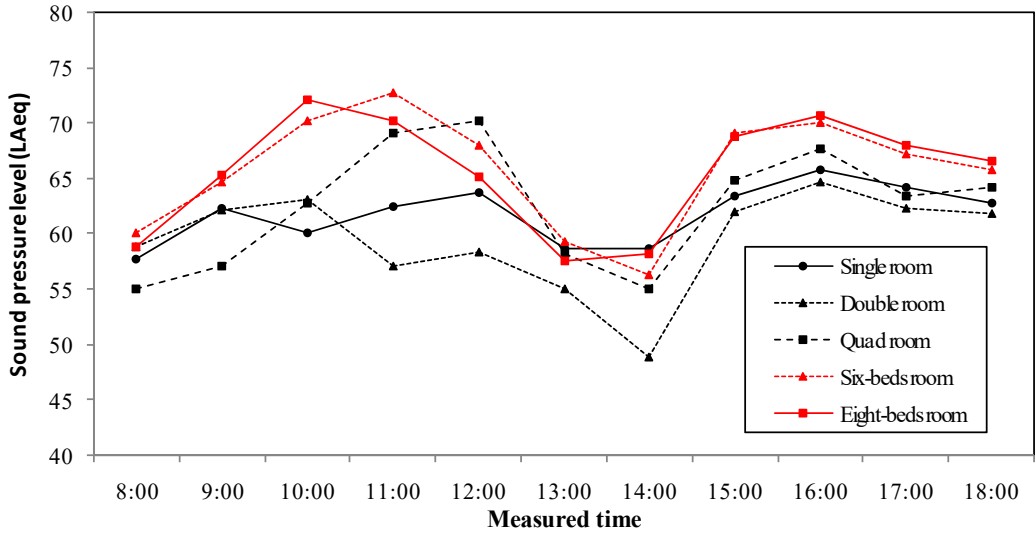

**Figure 3.** Measured LAeq over the test period with different ward capacities.

**Person-related factors**. Gender and age are factors that must be taken into account in the assessment of comfort in a hospital [41]. The mean difference between males and females in the evaluation of thermal and acoustic comfort was determined, and there is no significant difference ($p < 0.1$) between males and females in terms of acoustic comfort, with a similar mean value (female 4.21 and male 4.12), as shown in Figure 4. These results were consistent with those of previous studies, which suggested that the effect of gender on sound annoyance evaluation is generally insignificant [42–44]. There are significant gender differences in thermal comfort, as shown in Figure 5. Females are more critical than males about temperature, and females prefer higher room temperatures than males.

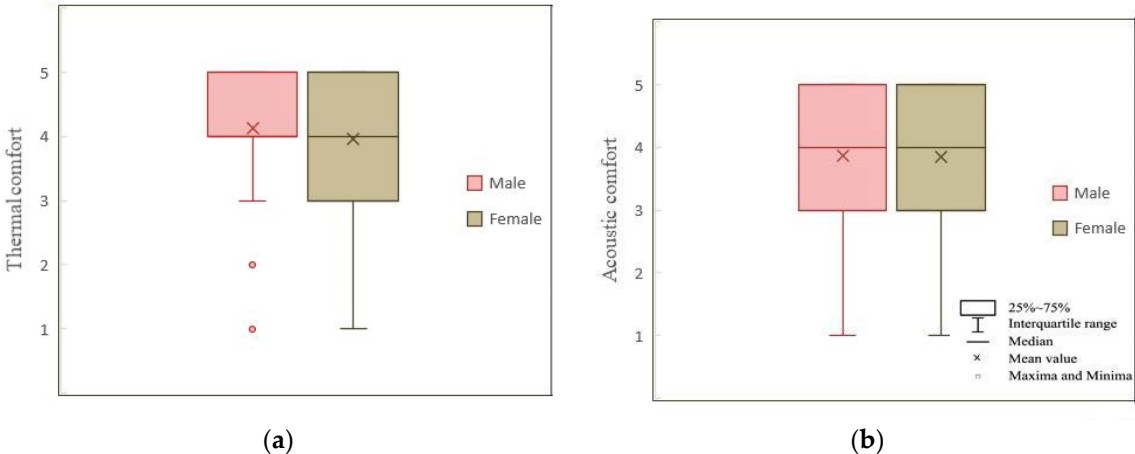

**Figure 4.** The gender difference on thermal and acoustic comfort. (**a**) Thermal, (**b**) acoustic.

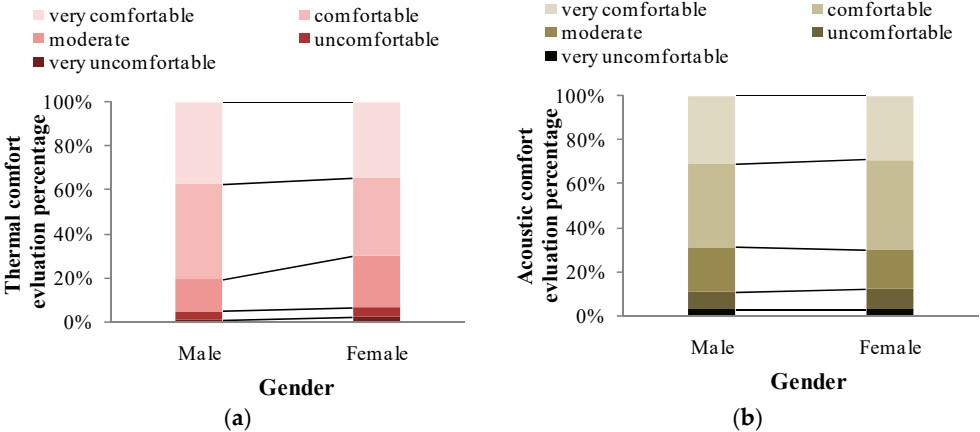

**Figure 5.** The percentage of comfort evaluation for different genders. (**a**) Thermal, (**b**) acoustic.

The age difference was significant ($p < 0.01$ or $p < 0.05$) for both thermal and acoustic comfort, as shown in Figure 6. Acoustic comfort was higher for older patients, and this is the opposite result to that found in other studies [45,46]. The possible reason for this is that the hearing level of the elderly is different from that of the young. Younger people were found to be highly annoyed by noise, which are the same results as those found in previous studies [47]. A hot and dry environment was considered to be more acceptable among elders than among young patients as shown in Figure 7. A previous study drew the opposite conclusion, namely, that thermal comfort was neither influenced by gender nor age [48].

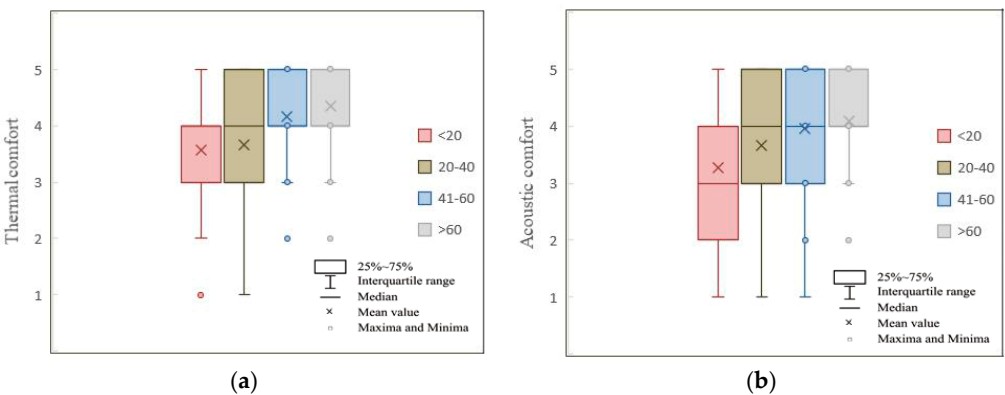

**Figure 6.** The age difference on thermal and acoustic comfort. (**a**) Thermal, (**b**) acoustic.

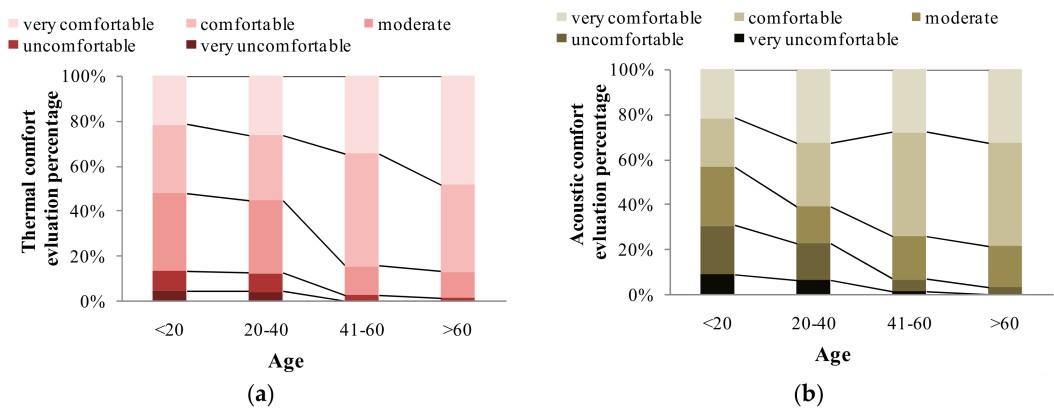

**Figure 7.** The percentage of comfort evaluation for different ages. (**a**) Thermal, (**b**) acoustic.

It is interesting to note that the duration of stay has a significant negative effect on thermal and acoustic comfort evaluation. As the stay length increased, the patients gave more negative comments, as shown in Figure 8.

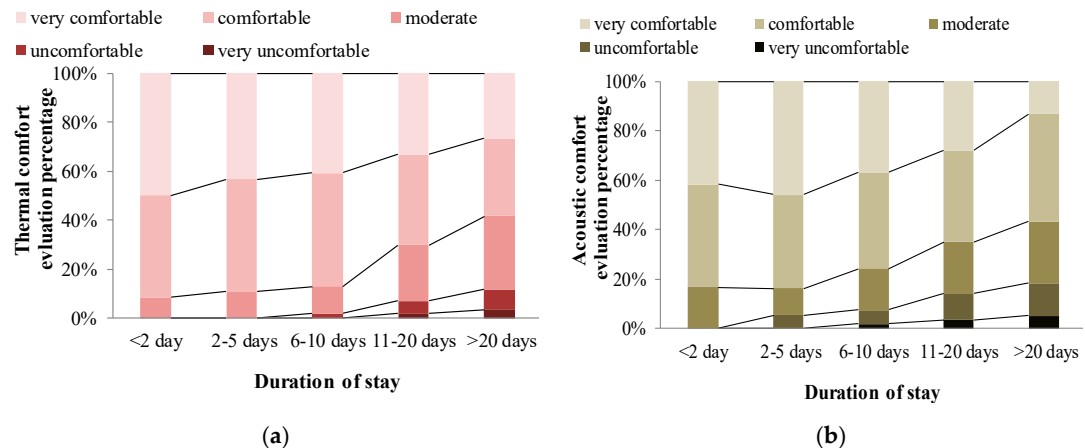

**Figure 8.** The percentage of comfort evaluation for different durations of stay. (**a**) Thermal, (**b**) acoustic.

## 5. Conclusions

This study investigated the interaction between sound and thermal comfort in hospital wards in China's heating region. An objective investigation was carried out, where participants evaluated sound comfort, thermal comfort, and overall comfort. Meanwhile, a subjective measurement of the sound pressure level, temperature, and humidity were carried out.

For overall comfort, the effects of sound and temperature are stronger than the effect of humidity, while the effects of sound and temperature are almost equal. For thermal comfort, the temperature in the wards was satisfactory, with a measured value of 26–30 °C, and highly acceptable at 27–29 °C. The patients had a preference for temperatures that are higher than the standard value in ISO 7730 and ASHRAE 55-2004. The relative humidity was unsatisfactory, and heating exacerbates this phenomenon. For acoustic comfort, the LAeq in the wards was satisfactory, with a measured value of 45–65 dB, and highly acceptable at 45–55 dB. There was a positive correlation between the acoustic comfort evaluation of the speech of staff, shouting, crying, and pager sounds, with shouting and crying being the most relevant.

The influence of thermal factors on sound evaluation showed that a low temperature has little effect on the evaluation of acoustic comfort, whereas for any type or volume of sounds, the higher the temperature, the more negative the evaluation. Irrespective of the type of sound source, when the humidity was low, the effect on acoustic comfort was the most obvious, but when the humidity was

medium or high, it was non-significant. In terms of shouting, crying, and pager sounds, the effects of temperature and humidity were more obvious, and high temperature and low humidity have a negative influence. With a medium temperature and humidity, the effect of improving acoustic comfort got better.

The similarities and differences in the evaluations of the sound and thermal influences also showed that there is an analogous trend in patients' environment comfort and preferences because of their high correlation coefficient. A satisfactory thermal environment can improve the evaluation of acoustic comfort, while an unsatisfactory thermal environment has the opposite effect.

Notwithstanding its limitation, this study clearly indicates that the special climatic environment in cold regions of China can cause changes in patient satisfaction. The building environments that affect patients' satisfaction include the thermal and acoustic environment. It should be noted that this study was focused specifically on patients in the cold region of China, and this specificity could affect the generalization of their findings of other patients. The geographical limitations of this study should be considered in interpreting these findings. Other regions or countries may produce different results. The three influencing factors selected in this study are dominated by thermal and acoustic comfort. Because of the dominant role of vision and air quality, future research can add lighting factors and indoor air quality as variables to study the influence of these additional factors on patient satisfaction.

**Author Contributions:** Conceptualization, Y.W., Q.M., L.L., and J.M.; Methodology, Y.W., Q.M., L.L., and J.M.; Investigation, Y.W.; Formal Analysis, Y.W.; Data Curation, L.L.; Writing-Original Draft Preparation, Y.W.; Writing-Review and Editing, Y.W., Q.M., L.L., and J.M.; Visualization, Y.W.; Supervision, Q.M. and J.M.; Project Administration, Q.M. and J.M.; Funding Acquisition, Y.W. and Q.M.

**Funding:** This research was supported by the National Natural Science Foundation of China (NSFC) (51808160, 51878210).

**Conflicts of Interest:** The authors declare no conflict of interest.

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
