# Peer review of "Interaction between Sound and Thermal Influences on Patient Comfort in the Hospitals of China’s Northern Heating Region"

_applsci, doi:10.3390/app9245551_

Round 1
Reviewer 1 Report
The authors present a questionnaire-based study for the evaluation of comfort in hospitals in relation to the indoor environment temperature, humidity, and sound pressure level. The authors also claim to have studied the interaction between all the considered factors and the resulting effects on the patient comfort.
From a general point of view, the manuscript is mostly focused on the analysis of the indoor thermal comfort, while the influence of the acoustic factors is overlooked. Since it has been proposed for a special issue in "Advances in Room Acoustics of Non-performing Public Spaces", I think that the it is slightly out of focus.
The presented results on the effects of the different thermal, humidity, and acoustic factors on the patient comfort are interesting, but not completely novel. Nonetheless, the accordance with the literature highlights the soundness of the presented results.
The most novel part of the manuscript, which would be the evaluation of the interaction of the different factors in the overall patient comfort, is extremely brief and not clear. The results of the statistical analysis are not presented e.g. in tabular form, reducing even more the clarity of the presented results.
Moreover, the Discussion section is completely missing from the paper. This hinders the comprehension and the importance of the presented results.
For those reasons, I do not consider the presented work suitable for publication in this form. However, since the results are potentially interesting, it might become suitable for publication after a major revision.
First of all, please change the title including all the factor you are considering, i.e. temperature, humidity, and pressure level.
Second: re-focus the manuscript to better match the scope of the special issue.
Third: extend the statistical analysis on the interaction between the three main factors (Section 3.4). For this purpose, a collection of the data in tabular form will improve the presentation effectiveness. Graphs and plots might also help.
Finally, add a comprehensive Discussion section, highlighting the importance of the presented results.
Other minor corrections:
1) Avoid using roman numerals for referencing to the bibliography. Please use western arabic numerals.
2) Mind the numbering of the sections, e.g. both Introduction and Methodology are numbered with "1".
3) In Page 2:
"China’s heating region is the annual average daily temperature stable ≤5℃ over than 90 days."
This sentence has no sense, probably it is incomplete. Please check it.
4) In Page 3:
"To measure the sound environment, the equivalent continuous A-weighted sound pressure level (LAeq) was immediately recorded using an 801 sound-level meter"
The model of the sound level meter is wrong (BSWA801). Moreover, you should describe the setup in more detail.
5) In Page 3:
"One measurement was performed every 10 s. The data for each location were recorded for 5 min. A mean value was calculated to obtain the corresponding LAeq."
The measurement procedure is not clear. As written, it seems that you just measure the acoustic pressure for 5min a day. Please describe the procedure with more detail.
6) In Figure 2: please align the time stamp to the graph ticks.
7) The writter english can definetly be improved in all the manuscript.
Reviewer 2 Report
Generally good and interesting article - and actually very interesting and very concluding results.
Abstract good and clear, but could possibly be shortened a bit.
Some phrases or some elements repeated in the Introduction.
Some numbering problems: 1. Introduction and 1. Methodology etc.
Same with literature citations, some problems with how those are presented.
And sometimes numbers, for example top of page 5 makes reference to Table 1 where it should read Table 3.
Otherwise description of research clear and understandable. One exception is sound level measurements: here it is stated:
"One measurement was performed every 10 s. The data for each location were recorded for 5 min. A mean value was calculated to obtain the corresponding LAeq xxvi."
This does not make sense or is not clear: measurements every 10s from 8:00 to 18:00, then averaged - and plotted as in figure 2?
Figure 2 on page 7 is clear, description is not clear.
Very good and clear results, especially Figure 1 and Figure 3!
These main results should be clearly stated (more clearly than currently) in the conclusion!
On question of age, text states that females prefer higher temperatures than males, and that elderly people prefer higher temperatures as well. This is not clear in Figure 4. Format of Figures 1 and 3 is much better (with different temperatures and sound levels on x-axis) than format of Figure 4, which shows how "critical" different populations are (especially 4f!) but not which conditions different populations prefer. This should be made clearer.
Reviewer 3 Report
The results presented in the paper, although very site-specific, are of interest. There are some part of the text that should be better explained or rephrased for clarity, as highlighted in the attached comments.

Reviewer 4 Report
The paper deals with an interesting and important topic: investigations into patients’ comfort due to environmental factors in care facilities are of utmost priority for society.
I have a few points I recommend the authors to address.
GENERAL COMMENTS
English should definitely revised by a native speaker, as there are several typos and many sentences are not smooth to read. This is crucial in my opinion: in the current version the English is not acceptable.
I think the literature review could be slightly expanded to better position this paper; in particular, considering that the special issue where this paper has been submitted is about acoustics, I would probably expand the acoustic bibliography; for some reviews and/or as a starting point, please have a look at (Thomas, et al., 2020) (Xie & Kang, 2012). Also, have a look at (Torresin, Albatici, Aletta, Babich, & Kang, 2019) for the review of methods for perception of indoor acoustic comfort in combination with other environmental factors.
Thomas, P., Aletta, F., Filipan, K., Vander Mynsbrugge, T., De Geetere, L., Dijckmans, A., . . . Devos, P. (2020). Noise environments in nursing homes: An overview of the literature and a case study in Flanders with quantitative and qualitative methods. Applied Acoustics, 159, 107103. doi:https://doi.org/10.1016/j.apacoust.2019.107103
Torresin, S., Albatici, R., Aletta, F., Babich, F., & Kang, J. (2019). Assessment Methods and Factors Determining Positive Indoor Soundscapes in Residential Buildings: A Systematic Review. Sustainability, 11(19), 5290. doi:10.3390/su11195290
Xie, H., & Kang, J. (2012). The acoustic environment of intensive care wards based on long period nocturnal measurements. Noise & Health, 14(60), 230-236.
The description of the measurements for the acoustic part should be reviewed by an experienced researcher as definitions of parameters/units and terminologies are not very accurate.
I am not sure the study does not require ethical approval in accordance with Declaration of Helsinki, the authors might need to read the declaration more carefully: please revise accordingly.
Section 1.1 (and wherever statistical approach is mentioned) – considering the categorical data of the questionnaire, Pearson is most probably not the right statistical test to use. Please re-run the statistical analysis with adequate methods.
In Figure 4, some strange patterns appear for both thermal and acoustic comfort when it comes to age difference. These should be more elaborated upon by the authors.
My main concern is that the statistical approach to this work and its datasets should be carefully re-thought and re-worked, because as it stands now, I think it is difficult to claim clear interaction effects and/or excluding other confounding factors.
MINOR COMMENTS
Title not very clear, try to make it more descriptive and/or rephrase it.
Abstract too general, should be more results-oriented with quantifiable information.
8:00am-18:00pm should be 08:00am-06:00pm
Figure 1: termal, should be thermal
Figure 4: durationg, should be duration
Round 2
Reviewer 1 Report
The authors presented a revisioned version of their manuscript.
The manuscript has been edited according to the suggestions and can now be considered for publication. I suggest, however, a few minor improvements:
Align the time stamps to the ticks in Figure 3 Section 3.2.1 is very interesting and could be extended to better describe the results reported in Table 5Author Response
Please see the attachment.

Reviewer 4 Report
I think the authors have provided sufficient amendments for the work to be endorsed.
Author Response
Thank you for giving us the opportunity to resubmit our revised paper. We greatly appreciate the reviewers’ helpful comments and suggestions for correcting our manuscript.